Improving classification based on physical surface tension-neural net for the prediction of psychosocial-risk level in public school teachers

Mosquera Navarro Rodolfo rmosqueran@usbcali.edu.co 1 2
Castrillón Omar Danilo 1
Parra Osorio Liliana 3
Oliveira Tiago 4
Novais Paulo 5
Valencia José Fernando 6
1 Departamento de Ingeniería Industrial, Universidad Nacional de Colombia , Manizales , Caldas , Colombia
2 Grupo Nuevas tecnologías trabajo y gestión, Universidad de San Buenaventura - Cali , Cali , Valle del Cauca , Colombia
3 Centro de Investigaciones Socio jurídicas, Facultad de Derecho, Universidad Libre , Bogotá , Cundinamarca , Colombia
4 Algoritmi Center, Universidade do Minho , Minho , Braga , Portugal
5 Department of Informatics/Algoritmi Center, Universidade do Minho , Minho , Braga , Portugal
6 Department of Ciencias y Tecnologías de la Información, Universidad de San Buenaventura - Cali , Cali , Valle del Cauca , Colombia
Laskey Kathryn
Electronic publication date: 2021 May 26
Publication date: 2021
Volume: 7
Electronic Location ID: e511
Received 2020 Jun 29; Accepted 2021 Apr 6
Copyright: ©2021 Mosquera Navarro et al.
Copyright year: 2021
Copyright holder: Mosquera Navarro et al.
License: This is an open access article distributed under the terms of the Creative Commons Attribution License, which permits unrestricted use, distribution, reproduction and adaptation in any medium and for any purpose provided that it is properly attributed. For attribution, the original author(s), title, publication source (PeerJ Computer Science) and either DOI or URL of the article must be cited.
License URL: https://creativecommons.org/licenses/by/4.0/

Keywords: Classification, Artificial Intelligence, Neural network, Physical surface tension-neural net, Psychosocial risk, State-School teachers

Funding: Convocatoria Nacional para el Apoyo al Desarrollo de Tesis de Posgrado o de Trabajos Finales de Especialidades en el área de la Salud de la Universidad Nacional de Colombia 2017-2018 Universidad de San Buenaventura-Cali-Facultad de Ingeniería-Nuevas tecnologías trabajo y gestión The authors received financial support from the “Convocatoria Nacional para el Apoyo al Desarrollo de Tesis de Posgrado o de Trabajos Finales de Especialidades en el área de la Salud de la Universidad Nacional de Colombia 2017-2018” via Resolution 21 of december 2017 from Vicerectoria of Investigaciones, who selected the proposal research: “Sistema de clasificación basado en técnicas inteligentes para identificar el grado de riesgo psicosocial en docentes de educación básica primaria y secundaria en colegios públicos de Colombia”, with identification number Hermes 40976 and Quipu code 201010016754. The authors also received financial support from the “Universidad de San Buenaventura-Cali-Facultad de Ingeniería-Nuevas tecnologías trabajo y gestión”. The funders had no role in study design, data collection and analysis, decision to publish, or preparation of the manuscript.

==============================
Background

Psychosocial risks, also present in educational processes, are stress factors particularly critical in state-schools, affecting the efficacy, stress, and job satisfaction of the teachers. This study proposes an intelligent algorithm to improve the prediction of psychosocial risk, as a tool for the generation of health and risk prevention assistance programs.

Methods

The proposed approach, Physical Surface Tension-Neural Net (PST-NN), applied the theory of superficial tension in liquids to an artificial neural network (ANN), in order to model four risk levels (low, medium, high and very high psychosocial risk). The model was trained and tested using the results of tests for measurement of the psychosocial risk levels of 5,443 teachers. Psychosocial, and also physiological and musculoskeletal symptoms, factors were included as inputs of the model. The classification efficiency of the PST-NN approach was evaluated by using the sensitivity, specificity, accuracy and ROC curve metrics, and compared against other techniques as the Decision Tree model, Naïve Bayes, ANN, Support Vector Machines, Robust Linear Regression and the Logistic Regression Model.

Results

The modification of the ANN model, by the adaptation of a layer that includes concepts related to the theory of physical surface tension, improved the separation of the subjects according to the risk level group, as a function of the mass and perimeter outputs. Indeed, the PST-NN model showed better performance to classify psychosocial risk level on state-school teachers than the linear, probabilistic and logistic models included in this study, obtaining an average accuracy value of 97.31%.

Conclusions

The introduction of physical models, such as the physical surface tension, can improve the classification performance of ANN. Particularly, the PST-NN model can be used to predict and classify psychosocial risk levels among state-school teachers at work. This model could help to early identification of psychosocial risk and to the development of programs to prevent it.

Introduction

Psychosocial risks are stress factors that can alter and unbalance a person’s resources and abilities to manage and respond to a flow of work activity, negatively affecting physical and psychological health (Sauter & Murphy, 1984). Among initial prediction models that enable identification of risks associated with work-related stress (Karasek, 1979) and workplace variables, some are based on mental stress resulting from workplace demands and decision-making. Workplace variables may cause the worker to feel their effort is inadequate, in accordance with the compensation obtained therefrom, and contribute to the development of work-related stress (Siegrist, 1996).

This situation is particularly critical in state-schools teachers, where work-related stress are present in educational processes (Kinman, 2001). A previous study (Collie, Shapka & Perry, 2012) shows how teachers’ perception of their work environment influence levels of: teaching efficacy, stress, and job satisfaction. The study demonstrates that the teachers’ perceptions of students’ motivation and behavior have the highest risk level. Workplace variables directly impact the perception of well-being among participants. Stress is negatively associated with the teaching efficacy variable. Additionally, stress influences directly on sense of job satisfaction, workload, and teaching efficiency. Indeed, prediction of psychosocial risk levels in state-school teachers is fundamentally important as a tool for the generation of health and risk prevention assistance programs.

Similar studies, focused on population different from teachers, have used machine learning techniques as Dynamic Bayesian Networks (DBN), Logistic Regression, Support Vector Machine, and Naïve Bayes classifiers (Liao et al., 2005; Subhani et al., 2017), to attempt a recognition of the patterns associated with workplace stress and for the detection of mental stress at two or multiple levels. Variables as facial expressions, eye movements, physiological conditions, behavioral data from activities in which users interact with the computer, and performance measurement, have been considered in that previous studies. A high level of stress is associated with symptoms including rapid heartbeat, rapid breathing, increased sweating, cold skin, cold hands and feet, nausea, and tense muscles, among others. Accuracy of 94.6% for two-level identification of stress and 83.4% accuracy for multiple level identifications have been reported.

Artificial neural networks (ANN) are a classification technique that in recent years have regained importance thanks to improvements associated with technology, as the deep learning (Saidatul et al., 2011; Sali et al., 2013). One of the crucial components of deep learning are the neural network activation functions, which are mathematical equations that determine the output of the ANN model, its accuracy, and also the computational efficiency. Different linear and nonlinear activation functions have been proposed in the literature (Tzeng, Hsieh & Lin, 2004), each one with its advantages and disadvantages, but reporting a better performance when nonlinear mathematical equations are included. The present work introduces a novel approach based on a modification in the activation function of the neural network, based on the theory of surface tension, in order to optimize the convergence of solutions in the classification processes. Indeed, the neural network calculates the desired output, using the surface tension function instead of the sigmoid function. In terms of mass and perimeter, these two surface tension equation parameters intervene to replace the network sigmoid function, with the aim to reduce data variability and dimensionality, and to improve the response in the classification and resulting error.

In the present study, the development of an new approach of neural network, based on Physical Surface Tension (Jasper, 1972) to model and predict psychosocial risk levels among Colombian state-school teachers, is presented. The Physical Surface Tension-Neural Net (PST-NN) approach is applied to psychosocial factors, musculoskeletal and physiological variables, present in academic environments in state-schools, in order to recognize their patterns, and thereby predict the type of risk to which a new individual may be exposed in such a work environment.

The next part of the document is organized as follows: first, the database, the preprocessing of the data, the definition of the new PST-NN approach, and the applied statistical tests, are described in ‘Materials & Methods’ section; then, the ‘Results’ section contains information about the training and test of the PST-NN approach, and its comparison with other published techniques; finally, the results are discussed and concluded in ‘Discussion’ and ‘Conclusions’ sections.

Materials & Methods

In this study, the results of tests for measurement of the psychosocial risk levels of 5,443 teachers, in five Colombian state-schools in cities in the same area, were analyzed. The data were obtained over a period of one and a half years. The dataset is a self-administered survey by labor psychologist and it was approved by university ethics committee public health at Universidad Nacional de Colombia, campus Manizales (Acta 01, SFIA-0038-17, legal document Mz. ACIOL-009-17, January 18, 2017). The dataset can be consulted in https://zenodo.org/record/1298610 (Mosquera, Castrillón Gómez & Parra-Osorio, 2018).

Database and data pre-processing

The dataset contains information about the following variables: (i) psychosocial; (ii) physiological, and; (iii) variables associated with pain and musculoskeletal disorders. Psychosocial risk factors may be separated into two main classes: those which have negative effects on health, and those which contribute positively to the worker’s well-being. Although both are present in all work environments, the present study considered those which negatively affect health in academic public-schools organizations (El-Batawi, 1988; Bruhn & Frick, 2011; Lippel & Quinlan, 2011; Weissbrodt & Giauque, 2017; Dediu, Leka & Jain, 2018).

Among the risk factor variables associated with work environment analysis, there was a total of 131 input variables: Xij = (psychosocial factors, j = 1, …, 123), Pij = (physiological factors, j =1, … 3) and Mij = (musculoskeletal symptoms, j = 1, … 5), where, i is the subject under study. Output variables were identified as the level of risk in which the person may be characterized Eij = Class [low risk (E1), medium risk (E2), high risk (E3), and very high risk (E4)]. Surface electromyography was performed to corroborate the musculoskeletal problems declared in patients with level of risk medium, high and very high, and confirmed in their clinical history. Electromyography data were collected with a BITalino (2017) (r) evolution Plugged kit (PLUX Wireless Biosignals S.A, Lisbon, Portugal) and validated by a medical specialist to find out if the patients actually had osteomuscular problems.

Table 1 Variables for intralaboral psychosocial risk factors. Adapted from: (Villalobosal et al., 2010).

Psychosocial risk variables	
Factor	Stressor	Description	
Leadership and social relations at work (L)	Leadership characteristics (L1)	Attributes of immediate superiors’ management, as related to task planning and assignment, attainment of results, conflict resolution, participation, motivation, support, interaction, and communication with employees.	
	Performance feedback (L2)	Information that a worker receives regarding the way in which they do their job. This information allows the identification of strengths and weaknesses, as well as action for performance maintenance or improvement.	
Control over work (C)	Clarity in the functions and role (C1)	Definition and communication of the role that the worker is expected to play within the organization, specifically as relates to work objectives, functions, results, degree of autonomy, and the impact of said role within the company.	
	staff training (C2)	Induction activities, training, and instruction provided by the organization, so as to develop and strengthen worker knowledge and abilities.	
	Skills and knowledge opportunities for its use and development (C3)	The possibility that a job provides an individual to apply, learn, and develop their abilities and knowledge.	
Work Demands (D)	Environmental demands and physical effort (D1)	Physical (noise, lighting, temperature, ventilation), chemical, or biological (viruses, bacteria, fungi, or animals) conditions, workstation design, cleanliness (order and sanitation), physical loads, and industrial security.	
	Emotional demands (D2)	Emotional demands
Require worker ability to:
(a) Understand the situations and feelings of others, and (b) exercise self-control over their own emotions or feelings, in order to avoid affecting work performance.	
	Quantitative demands (D3)	Demands relative to the amount of work to be performed and the time available to do so.	
	Influence of work on the non-work environment(D4)	Work demands on individuals’ time and effort which impact their after-work activities, personal, or family life.	
	Mental workload demands (D5)	These refer to the cognitive processing demands required for a task, and which involve superior mental attention, memory, or information analysis processes to generate a response.
The mental load is determined by the information characteristics (quantity, complexity, and detail), as well as the time available to process said load.	
	Working day demands (D6)	Work time demands made on an individual, in terms of duration and work hours, including times for pauses or periodic breaks.	
Rewards (R)	Work rewards (R1)	Remuneration granted to the worker to compensate their effort at work. This remuneration includes recognition, payment, and access to wellness services and possibilities for growth.	

Redundant psychosocial factors (Xij) were filtered by means of rank importance of predictors using ReliefF algorithm procedure (1) (Robnik-Š & Kononenko, 2003), with the goal of identifying noisy variables in the dataset using the Chebyshev metric criteria. The ReliefF algorithm located important predictors throughout the 10 nearest neighbors and put the 123 Xij independent factors into groups. Predictor numbers were listed by ranking, and the algorithm selected the predictors of greatest importance. The weights yielded weight values in the same order as that of the predictors. Distances between factor pairs, at this weight, were measured once again, and the factor with the lowest total value (distance) was chosen, which yielded 12 Xij factors per group. It further added physiological (Pij) and musculoskeletal symptom variables (Mij). The algorithm recognized the variables with the lowest value and punished those predictors (risk associated with each individual Xir, where r = 1, …, 4 represents the risk level: low risk (Xi1), medium risk (Xi2), high risk (Xi3), and very high risk (Xi4)), which produced different values for neighbors in the same group (risk factors group Fij), and it increased those which produce different values for neighbors in different groups. ReliefF initially defined predictor weights Rij at 0, and the algorithm subsequently selected a random value Xir, iteratively. The k-nearest values Xir for each group were identified, and all predictor weights Fij for the nearest neighbors Xiq were updated (Robnik-Šeck & Kononenko, 2003, p. 26). (1) WA:=WA−∑j=1kdiffA,Ri,Hjm.k+∑C≠classRi+PC1−PclassRi ∑j=1kdiffA,Ri,MjCm.k

Where,

Ri is randomly selected instances.

Hi is k nearest hits (k-nn with the same class). Mj(C) is k nearest misses (k-nn with the different class).

WA is the quality estimation for all attributes A for Ri, and Hj and misses values Mj(C).

1−PclassRi is the sum of probabilities for the misses classes.

m is the processing time repeated.

In total, 20 input variables Eij = Xij + Pij + Mij were selected (Tables 1–3): twelve variables Xij = (j = 1, …, 12), which constituted psychosocial variables; three physiological variables Pij = (j = 1, …, 3), and; five variables associated with musculoskeletal symptoms Mij = (j = 1, …, 5). This variables were normalized, in accordance with Eq. (2). (2) Enormalized=E−EminEmax−Emin

Table 2 Physiological variables.

Physiological variables	
Heart rate (P1)	Heart rate is the speed of the heartbeat measured by the number of contractions (beats) of the heart per minute (bpm).	
Electrodermal activity (P2)	Property of the human body that causes continuous variation in the electrical characteristics of the skin. Skin conductance can be a measure of emotional and sympathetic responses.	
Electromyography (P3)	Is an electrodiagnostic medicine technique for evaluating and recording the electrical activity produced by skeletal muscles.	

Table 3 Musculoskeletal symptoms.

Physiological variables (Related to work absenteeism and psychosocial factors)	
Symptoms	Description	
Headache & (M1)
Cervical pain	A headache in general is a sign of stress or emotional distress, and can be associate to migraine or high blood pressure, anxiety or depression. Some patients experience headache for 2 hours or less. (Headache Classification Committee of the International Headache Society (IHS), 2013).	
Migraine (M2)	Migraines can be associate to by a severe headache that often appears on one side of the head. They tend to affect people aged 15 to 55 years. Symptoms include hyperactivity, hypoactivity, depression, fatigue and neck stiffness and/or severe pain (Headache Classification Committee of the International Headache Society (IHS), 2013).	
Shoulder pain (M3)	The pain is elicited or aggravated by movement. Pain and stiffness usually restrict the use of the Superior limbs and thereby limit daily activities during work vanderHeijden1999.	
Arm pain (M4)	Arm pain is caused by repetitive movements at work, usually the symptoms are described as pain, discomfort, or stiffness that occurs anywhere from your shoulders to your fingers.	
Back pain (M5)	Back pain at work usually can affect people of any age, heavy lifting, repetitive movements and sitting at a desk all day can produce a injury.	

Where, E corresponds to the variable to be normalized, Emax is the maximum value of each variable, Emin is the minimum value, and Enormalized is the normalized variable within the −1 to 1 range.

Basis of the surface tension-neural net algorithm (PST-NN)

The approach was based on the theory of liquid surface tension (Macleod, 1923; Jasper, 1972; Tyson & Miller, 1977), given by Eq. (3). Liquid surface tension is defined as the amount of energy necessary to increase surface area of a liquid per unit of area. Surface tension (a manifestation of liquid intermolecular forces) is the force that tangentially acts, per unit of longitude, on the border of a free surface of a liquid in equilibrium, and which tends to contract the said surface (Adamson & Gast, 1967a). The cohesive forces between liquid molecules are responsible for a phenomenon known as surface tension (Fowkes, 1962; Adamson & Gast, 1967b; Tida & Guthrie, 1993; Law, Zhao & Strojnisìtva, 2016; Almeida et al., 2016).

(3) γ=F2L

Where, γ is the surface tension that measures the force per unit length (in the model γ is the classification risk level), F is the force required to stop the side from starting to slide, L the length of the movable side, and the reason for the 1/2 is that the film has two surfaces (Macleod, 1923). In this model, the multiplication of the perimeter of an object by the surface tension of a liquid yields the force that a liquid exerts on its surface, on an object, in order to prevent said tension from breaking. As such, if the weight of an object is greater than the force exerted by the liquid on its surface, the object tends to sink.

The theory of surface tension addresses cohesion between molecules in a liquid, and their energetic relationship with the exterior, generally a gas. When submitted to a force that breaks the energetic state of molecular cohesion, the surface of a liquid causes the object producing internal force in the liquid to sink. This proposal sought to emulate the surface tension theory in the psychosocial analysis of risk factors present in work environments and their degrees of risk, from the viewpoint of improving a machine learning model. It used and adapted the said theory to improve risk classification and modify the necessary parameters of a neural network (the number of layers, nodes, weights, and thresholds) to reduce data dimensionality, and increase precision.

Implementation of the PST-NN algorithm

variables Eij became two physical variables, perimeter and mass, throughout an artificial neural network with four layers. Three of these layers constitute the architecture of a standard neural network, with the difference that, the last level contains a new neural network model based on physical surface tension (Adamson & Gast, 1967b). Eighty neurons were used in layers one and two, due to the fact that substantial changes were not registered using more neurons in these layers . Additionally, just two neurons were used for layer 3, in order to annex the new proposed surface tension layer. The architecture of the artificial neural classification network is shown in Fig. 1. This included the three standard neuron layers, as well as a fourth layer with a novel design.

Figure 1 Physical surface tension-neural net.

For the initialization of the neural network parameters, the Nguyen-Widrom algorithm was used (Pavelka & Prochazka, 2004; Andayani et al., 2017), in which random parameters were generated. However, the advantage of this was that the parameters distribute the active neural regions much more uniformly in layers, which improved neural network training, as it presented a lower learning rate error from the beginning.

Layer 1 output calculation: The 20 input variables of a specific individual from the training set, a vector called E, went through an initial layer of 80 neurons. Each neuron had 20 parameters, called weights, which multiplied each input variable of vector E. A parameter called bias b was added to this multiplication. It was associated with each neuron, which results in the partial output of Layer 1. This procedure is described throughout the following equation: (4) yk1=∑i=120Ei∗wk,i1+bk1fork=1to80

(5) y1=y11,y21…y801

Where Ei is the i variable of the individual chosen from the training set, wk,i1 is the k neuron’s weight in Layer 1, which is multiplied by variable i, bk1 is neuron k’s bias in Layer 1, which is added to the total, and yk1 is the result of each k neuron. These 80 results were represented by y1 vector, and y1 went through a hyperbolic tangent transfer function, as this is a continuous transfer function, and is recommended for pattern-recognition processes (Harrington, 1993).

Layer 1 output is described in the following equation (6) Yk1=21+e−2∗yk1−1fork=1to80

(7) Y1=Y11,Y21…Y801

Where, e is the exponential function and Y1 is the final output for Layer 1 and is composed of 80 outputs, one for each neuron.

Layer 2 output calculation: The 80 outputs from Layer 1, Y1, become the inputs for Layer 2, which presents the same number of neurons as Layer 1. As such, in accordance with the procedure performed in Layer 1, the following equations are obtained: (8) yk2=∑i=180Yi1∗wk,i2+bk2fork=1to80

(9) y2=y12,y22…y802

Where, Yi1 is the output of neuron i from Layer 1, wk,i2 is the weight of neuron k, associated with the output of neuron i in Layer 1, bk2 is neuron k’s bias in Layer 2, and y2 includes the 80 responses of each neuron, prior to passing through the transfer function. In order to obtain the final output for Layer 2 (Y2) the hyperbolic transfer function is applied: (10) Yk2=21+e−2∗yk2−1fork=1to80

(11) Y2=Y12,Y22…Y802

Layer 3 output calculation: The 80 outputs for Layer 2 were the inputs of Layer 3, which contains two neurons (12) Yk3=∑i=180Yi2∗wk,i3+bk3fork=1to2

(13) Y3=Y13,Y23=m,Per

Where Yi2 is the output of neuron i in Layer 2, wk,i3 is the weight of neuron k in Layer 3, which multiplies the output of neuron i in Layer 2, and Yk3 is the final output of each of the two neurons represented in vector Y3. In the approach of Physical Surface Tension Neural Net (PST-NN), these two output variables were then considered mass (m) and perimeter (Per), respectively, which went into a final layer called the surface tension layer. This was composed of four neurons, one neuron for each risk level. Each of these contributed to a balance of power defined by the following equation: (14) Ok=1−eF2Lfork=1to4

Where, (15) Ok=1−e−m∗gTk∗Perfork=1to4

(16) O=O1,O2,O3,O4

With: (17) Tk=22.1;47.7;72.8;425.41

Where m is the mass that corresponds to the output of the first neuron from Layer 3, g is the value of the gravity constant 9.8ms2) (The multiplication of mass times gravity m∗g yields the weight of an object); Per, the perimeter is the output of the second neuron, from Layer 3; and Tk is the value of the surface tension in neuron k, which were associated to the surface tensions of four liquids: Ethanol (22.1), Ethylene glycol (47.7), Water (72.8), and Mercury (425.41) (Surface tension value (mN/m) at 20 °C) (Jasper, 1972). The four liquids shown above were used, as they are common, relatively well-known, and present different surface tensions. Here, the main idea was the relationship that exists between the four surface tensions and the different weights of objects that can break the surface tension of the liquid. For our model, the surface tension of each liquid was similar to each level of psychosocial risk, where the lowest risk level corresponded to the surface tension of the ethanol, and the very high-risk level was equivalent to the surface tension of the mercury. In this sense, when a person has, according to the psychosocial evaluation, a high-risk level, the parameters in the new surface tension neuron will be equivalent to having traveled the surface tension of ethanol, of ethylene glycol, to finally break the surface tension of the Water. Theoretically, at this point the liquid tension will be broken and the classification of the patient under study will be high risk.

The Ok transfer function was used, owing to its behavior. Note that:

(18) limx→∞1−e−x=1

(19) limx→01−e−x=0

Thus, when the force exerted by the weight was greater than that exercised by the liquid, the surface tension was broken (See Fig. 2). When this occurs, Ok tends to be one, and when it does not, the value of Ok tends to be zero.

The correct outputs for the four types of risk must be as shown below: (20) Risk1,O=1,0,0,0Risk2,O=1,1,0,0Risk3,O=1,1,1,0Risk4,O=1,1,1,1

Risk 4 breaks through all surface tensions, while Risk 1 only breaks through the first surface tension.

Figure 2 Classification method based on physical surface tension.

Computation of the error backpropagation

The four outputs contained in O were compared to the response Eij, which the neuron network should have yielded, thus calculating the mean squared error: (21) errorcm= ∑k=14Ok−Ek22.

The following steps calculated the influence of each parameter on neuron network error, through error backpropagation, throughout partial derivatives. The equation below was derived from Ok:

The derivative of the error, regarding neural network output: (22) ∂errorcm∂Ok= ∑k=14Ok−Ek

The derivative of the error, regarding layer 3 output: (23) ∂errorcm∂Y13=∂errorcm∂m=∂errorcm∂Ok∗∂Ok∂m

(24) ∂Ok∂m=gTk∗Pere−m∗gTk∗Per

(25) ∂errorcm∂Y23=∂errorcm∂Per=∂errorcm∂Ok∗∂Ok∂Per

(26) ∂Ok∂Per=−m∗gTk∗Per2e−m∗gTk∗Per

(27) ∂errorcm∂Y3=∂errorcm∂Y13,∂errorcm∂Y23

Derivative of error, according to layer 3 weights: (28) ∂errorcm∂wk,i3=∂errorcm∂Yk3∗∂Yk3∂wk,i3fork=1to80,withi=1and2

(29) ∂Yk3∂wk,i3=Yi2

Derivative of error, according to layer 3 bias:

(30) ∂errorcmbk3=∂errorcm∂Yk3∗∂Yk3bk3

Derivative of error, according to layer 2 output: (31) ∂errorcm∂Yi2= ∑k=12∂errorcm∂Yk3∗∂Yk3Yi2fori=1to80

(32) ∂errorcm∂Yi2= ∑k=12∂errorcm∂Yk3∗wk,i3fori=1to80

Derivative of error, according to layer 2 weights: (33) ∂errorcm∂y2=∂errorcm∂Y2∗∂Y2∂y2

(34) ∂Y2∂y2=1−Y22

(35) ∂errorcmwk,i2=∂errorcm∂Y2∗∂Y2∂y2∗∂y2wk,i2fork,i=1to80

Derivative of error, according to layer 2 bias: (36) ∂errorcmbk2=∂errorcm∂Y2∗∂Y2∂y2∗∂y2bk2

(37) ∂y2bk2=1

Derivative of error, according to layer 1 output: (38) ∂errorcm∂Y1=∂errorcm∂Y2∗∂Y2∂y2∗∂y2∂Y1

Derivative of error, according to layer 1 weights:

(39) ∂y2∂Y1=wk,i3fori,k=1to80

(40) ∂errorcm∂y1=∂errorcm∂Y2∗∂Y2∂y2∗∂y2∂Y1∗∂Y1∂y1

(41) ∂Y1∂y1=1−Y12

(42) ∂errorcm∂wk,i1=∂errorcm∂Y2∗∂Y2∂y2∗∂y2∂Y1∗∂Y1∂y1∗∂y1∂wk,i1

(43) ∂y1∂wk,i1=Eifork=1to80;i=1to20

Derivative of error, according to layer 1 bias:

(44) ∂errorcm∂b1=∂errorcm∂Y2∗∂Y2∂y2∗∂y2∂Y1∗∂Y1∂y1∗∂y1∂b1

(45) ∂y1∂b1=1

(46) ∂errorcm∂parameters=∂errorcm∂b1,∂errorcm∂w1,…∂errorcm∂b2,∂errorcm∂w2,…∂errorcm∂b3,∂errorcm∂w3

The new parameters in iteration n+1 were calculated throughout the conjugate gradient method: (47) parametersn+1=parametersn+ηn∗dn

Where, (48) ηn∗dn

Depends on the (49) ∂errorcm∂parametersvalues.

This procedure was repeated, beginning at step in (4) for the remaining training data, thus completing the first iteration. Later, iterations were performed repeatedly until there was an artificial neural network convergence, according with the following three stop criteria: (a) Minimum performance gradient, the value of this minimum gradient is 10−6. This tolerance was assigned for adequate neuron network learning; (b) Performance, in order to measure neural network performance, the mean squared error was employed. The value to be achieved is zero, so as to avoid presenting neural output errors; (c) Number of Iterations, the training was stopped if 300 iterations were reached. A high number of iterations was chosen, as ideally, it stopped with error criteria.

The code developed in Matlab V9.4 software can be consulted here: https://codeocean.com/capsule/6532855/tree/v1 (Mosquera, Castrillón Gómez & Parra-Osorio, 2019).

Statistical analysis

The data set was divided into training (80%) and test (20%) groups (train/test split) as published in (Vabalas et al., 2019). For the evaluation of the algorithm the following metrics were used (Rose, 2018): (a) Sensitivity, which provides the probability that, given a positive observation, the neural network will classify it as positive (50); (b) Specificity, which provides the probability that, given a negative observation, the neural network will classify it as negative (51); (c) Accuracy, which gives the total neural network accuracy percentage (52) and, (d) the ROC curve by plotting the sensitivity (true-positive rate) against the false-positive rate (1 − specificity) at various threshold settings. Different authors in other studies as have been used the sensitivity, specificity, and, AUC, for the performance statistics within the independent dataset (Le, Ho & Ou, 2017; Do, Le & Le, 2020; Le et al., 2020).

(50) Sensitivity=TPTP+FN

(51) Specificity=TNTN+FP

(52) Accuracy=TP+TNTP+TN+FP+FN

Where TP, TN, FP and FN denote the number of true positives, true negatives, false positives and false negatives, respectively. In order to analyze the stability of the system in the results obtained, a variance analysis, using (53) was performed, to establish whether there were significant differences in the results. In this analysis, representing the response to the variables, Ti, was the effect caused by nth treatment, and εi, the nth experimental error. The information collected must comply with independence and normality requirements. The variance analysis was performed under a confidence interval of 99.5% (Rodriguez, 2007): (53) Yi=μ+Ti+εi

The efficiency of the PST-NN approach was compared with previous published techniques (Mosquera, Parra-Osorio & Castrillón, 2016; Mosquera, Castrillón & Parra, 2018a; Mosquera, Castrillón & Parra, 2018b; Mosquera, Castrillón & Parra-Osorio, 2019), which were applied over the original data included in the present work. Accuracy was the metric used to make the comparison between PST-NN and Decision Tree J48, Naïve Bayes, Artificial Neural Network, Support Vector Machine Linear, Hill Climbing-Support Vector Machine, K-Nearest Neighbors-Support Vector Machine, Robust Linear Regression, and Logistic Regression Models.

Results

Adjustment of the PST-NN approach

The 20 input variables (psychosocial, physiological, and musculoskeletal symptoms) belonging to the 5443 subjects were used to train and test the Physical Surface Tension Neural Net (PST-NN), according with the level of risk in which the person may be characterized (low, medium, high, and very high risk).

Figure 3 shows the mean squared error that was obtained during the training and testing process of the PST-NN approach, as a function of the iterations number that was used in the adjustment of the neural network parameters. The trend of the blue line, corresponding to the training group, showed how the mean squared error rapidly decreases around the first 100 iterations, reaching a plateau for higher values of the iterations. This plateau indicated that the neural net model has reached the parameters optimization and therefore, any additional increment in the number of iterations not significatively improve the parameters adjustment. Concretely, in in this study and for the next results, 108 iterations were considered in the adjustment of the PST-NN parameters. The curve of the mean squared error corresponding to the testing group (red line) showed a similar behavior to the training group. Indeed, the following results were reported only for the test set.

Figure 3 Iterations performance in the Physical Surface Tension-Neural Net model.

In relation with the layer that represents the surface tension model in the PST-NN approach (Fig. 1), Figure 4 showed the results of the perimeter and mass outputs for each subject in the test group, according with the risk level. The outputs were plotted in a XY graph, where the mass output corresponds to the X axes and the perimeter to the Y axes. As result, it was possible to see that the points were grouped in specific areas as a function of the risks level. In this sense, the types of risk may additionally be interpreted in physical form. Indeed, the highest risk in the graph corresponded to the red crosses, which present mass values which were relatively larger than the rest, along with relatively smaller perimeters, which cause the surface tension of the four liquids to break. The lowest risk (represented in blue with asterisks) had relatively high perimeters and relatively low masses, which cause them to remain on the surface of certain liquids.

Figure 4 Mass Vs perimeter classification risk model.

The square root of the mass/perimeter relationship was represented in Fig.5. This transformation of the relationship between mass and perimeter was applied only for improved visualization of the separations between the risk levels. The figure showed that the lowest value of the square root of the mass/perimeter relationship corresponded to the lowest risk level and the highest value to the highest risk level.

Classification performance of the PST-NN approach

The specific confusion matrix for the test set (Table 4) showed the performance of the PST-NN algorithm, as a function of the TP, TN, FP and FN. The number of subjects in each target risk group was 116, 117, 347, and 521 for risk levels 1, 2, 3, and 4, respectively. The number of subjects classified by the algorithm in each risk group was 109 (Risk Level 1), 113 (Risk Level 2), 339 (Risk Level 3), and 540 (Risk Level 4).

Table 4 The Confusion matrix for Physical Surface Tension-Neural Net model for the prediction of psychosocial risk level.

For test set (20%).

Confusion Matrix Test	
		Risk 1	Risk 2	Risk 3	Risk 4		
Output class	Risk 1	107
9.7%	1
0.1%	1
0.1%	0
0%	98.2%
1.8%	
Risk 2	3
0.3%	91
8.3%	19
1.7%	0
0.0%	80.5%
19.5%	
Risk 3	4
0.4%	13
1.2%	309
28.1%	13
1.2%	91.2%
8.8%	
Risk 4	2
0.2%	12
1.1%	18
1.6%	508
46.1%	94.1%
5.9%	
		92.2%
7.8%	77.8%
22.2%	89.0%
11.0%	97.5%
2.5%	92.2%
7.8%	
		Risk 1	Risk 2	Risk 3	Risk 4		
		Target Class		

Table 5 included the values of sensitivity, specificity, accuracy and AUC for each of the risk levels in the test set. The highest sensitivity value was 97.5% (Risk level 4) and the lowest sensitivity value was 77.8% (Risk level 2), indicating that Risk Level 2 was the most difficult type of risk to classify. On the contrary, the best specificity value was obtained in Risk level 1 (98.2%) and the lowest was in Risk level 3 (96.0%). In relation to the accuracy, Risk Level 2 had the lowest value, indicating that the surface tension neural network would correctly classify an individual, with a probability of the 82.7%, to belong or not to this risk level (it includes true positive and true negative cases). The risk levels with the greatest accuracy values were Risk level 1 followed by Risk level 4, with values of 98.85% and 97.37, respectively. Complementary, Figure 6 showed the receiver operating characteristic curves (ROC curve) for each risk level for the test set. Risk level 4 had the best classification with AUC value of 0.984 (Table 5), while Risk level 2 was the one that presents the most confusion on classification (AUC =0.883).

Table 5 Statistical measures for the classification test (20%) for the four risk levels.

	Risk Levels	
Statistical measure	Risk level 1	Risk level 2	Risk level 3	Risk level 4	
Sensitivity	92.2%	77.8%	89.0%	97.5%	
Specificity	98.2%	96.8%	96.0%	96.6%	
Accuracy	98.2%	82.7%	96.0%	97.3%	
AUC	0.961	0.883	0.971	0.984	

Figure 6 ROC Curve.

Finally, the performance of the PST-NN approach was compared in terms of accuracy against the results of linear, probabilistic, and logistic models, previously published (see Table 6). The proposed PST-NN method had the best accuracy value (97.31%), followed by Support Vector Machines (92.86%), Hill-Climbing-Support Vector Machines (92.86%), and Artificial Neural Networks (92.83%). The lowest accuracy values were obtained with the Robust Linear Regression (53.47%), and Logistic Regression (53.65%) techniques. The statistical stability analysis, based on the ANOVA method, showed statistically significant differences between PST-NN and the other techniques, in relation to the accuracy values, with p-value <0.05.

Table 6 Results applying different classification techniques in psychosocial factors dataset.

Id	Algorithm	% classification averagea	
1	J48	91.29	
2	Naïve Bayes	89.71	
3	ANN	92.83	
4	SVM	92.86	
5	HC-SVM	92.86	
6	SVM-RBF	89.26	
7	KNN-SVM	86.66	
8	Robust Linear Regression	53.47	
9	Logistic Regression	53.65	
10	Proposed Method: PST-NN	97.31	
Notes.

a Accuracy.

Discussion

In this study, the Physical Surface Tension-Neural Net (PST-NN) approach was developed and applied to model and predict psychosocial risk levels among Colombian state-school teachers. The fundamental point of the structure of this model was the improvement of the neural model by the adaptation of a layer that includes concepts related to the theory of physical surface tension. Indeed, the psychosocial risk level was associated with the probability that a “surface” can be broken as a function of the psychosocial, physiological, and musculoskeletal factors impact. For each risk level, a different value of the physical surface tension was set in analogy with the surface tensions of four common liquids (Ethanol, Ethylene glycol, Water, and Mercury). This attempts to benefit from the characteristics of neural networks and increase precision via innovation (theory of physical surface tension), in the form of neural network modification. It is expected that this combination enables the elimination of linear model deficiencies and the development of an approach to the real world, with fewer shortcomings.

This technique presented an important advantage, due it allowed the dimensionality in the input variables to be reduced. In this study, the 20 input variables in the first layer of the neural network were reduced to 2 variables (mass and perimeter) in the surface tension layer, in order to facilitate the classification process. In this layer, the surface tension equation intervened to replace the network sigmoid function, which reduced data variability and dimensionality, improving the response in the classification and resulting error. The results reported in Figs. 4 and 5 supported this behavior, so it was possible to see a clear grouping of the subjects according to the risk level group, as a function of the mass and perimeter outputs. This was according to the surface tension theory by which a low mass and high perimeter reduce the probability of breaking the surface, and on the contrary, a high mass with a low perimeter increases that probability.

Figure 5 Visualization of risk separations in the model.

The neural network models possess high flexibility and fewer parameters compared with parametric models (Darvishi et al., 2017). Results in Fig. 3 showed that the neural model iteration process quickly catches up to the number of iterations necessary to establish the model and provide objective, precise results. However, in supervised machine learning, overfitting could be present, indicating that model does not generalize well from observed data to unseen data (Ying, 2019). Because of that, the model performs better on training set that in testing set. In order to reduce the effect of overfitting during the adjustment process of the PST-NN parameters, the train/test split methodology (Vabalas et al., 2019), besides to the control in the number of iterations during the neural network training, and the normalization and reduction in dimensionality of the input data, were used (Ying, 2019). However, the number of subjects in each risk level group was not uniformly distributed, being the Risk level 4 the group with more subjects, and Risk level 1 and 2 the groups with less subjects. This situation could generate that the PST-NN model tends to memorize in more detail the data belonging to Risk level 4, and in less detail the data of Risk level 1 and 2.

The application of the PST-NN approach to the data belonging to Colombian state-school teachers, showed an average accuracy value of 97.31% in the psychosocial risk classification, including all the risk level groups and all the subjects in the database. The confusion matrix results (Table 4) and ROC curve (Fig. 6) demonstrated that the PST-NN model was highly efficient, in terms of psychosocial risk classification, as compared to other experiments and models (Larrabee et al., 2003; Baradaran, Ghadami & Malihi, 2008; Aliabadi, Farhadian & Darvishi, 2015; Farhadian, Aliabadi & Darvishi, 2015; Yigit & Shourabizadeh, 2017; Jebelli, Khalili & Lee, 2019). The level of precision and low error percentage of PST-NN approach demonstrated the ease adaptation of the mathematical structure to the input variables, generating a model that can be used to perform preventive interventions in occupational health by way of prediction, based on psychosocial, physiological, and musculoskeletal factors.

Psychosocial, physiological, and musculoskeletal factors fundamentally involve non-linear relationships. While neural networks are linear models that provide adequate approaches for the classification problem, the introduction of a physical concept to the neural model, such as the physical surface tension theory, adapted better to the type of data present in organizational and psychosocial climate evaluations. As such, the PST-NN model, by way of the transformation and neural suitability procedure, may discover improved solutions. Alternatively, other authors (Tzeng, Hsieh & Lin, 2004; Hong et al., 2005; Azadeh et al., 2015; Jebelli, Khalili & Lee, 2019) have avoided the non-linear relationships transforming the data in four linear variables: a positive relationship, negative relationship, no relationship, and non-linear relationship, in studies to analyze the performance and personnel turnover data. However, the results showed values of classification and prediction that could be improved.

The performance of the PST-NN approach, for psychosocial risk level prediction, showed better average accuracy value (97.31%) than the results of support vector machine linear models, neural networks, probabilistic models, linear and logistic regression models, and decision tree models, previously published (Table 6). Particularly, the ANN model, corresponding to a perceptron neural network without the modification proposed in this study, only reached an average accuracy value of 92.83%, suggesting that the modification introduced by the PST-NN approach could significatively improve the classification performance. The use of regression techniques showed that the misclassification probability was high, with accuracy values of 53.47% and 53.65% for the Robust Linear Regression and Logistic Regression, respectively. This suggest that linear models are not well fitted to the type of data that were used in the present study.

The results of previous experiments support the opinion that the strategy of combining different methods (physical surface tension theory and artificial neural networks) may improve predictive performance. Similar strategies have been applied previously to work safety and health classification problems, for work stress, psychosocial factor, and mental stress factor prediction (Jackofsky, Ferris & Breckenridge, 1986; Somers, 1999; Zorlu, 2012; Sriramprakash, Prasanna & Murthy, 2017; Subhani et al., 2017; Xia, Malik & Subhani, 2018; Lotfan et al., 2019).

In all industries and organizations, analysis of the psychosocial risk level is very important. Studies have shown the direct relationship between psychosocial risks and the gross domestic products of nations (Dollard et al., 2007). The implementation of artificial intelligence techniques can contribute to the development of this field of research, which could be called psychosocial analytics. It’s vital the development of these types of tools in global occupational and public health. Colombia’s leadership in Ibero-America in the development of tools which contribute to the occupational health and safety field is highlighted by this kind of work.

As a limitation, is important to point that the performance of the prediction model depends on both the quality and quantity of the dataset, as well as the optimal structure design. Indeed, and for the PST-NN model developed in this study, the performance will be affected by the psychosocial factor management, which depends, among other things, of the teacher population and if the data is taken by region, or similar geographical areas or annexes. When this is not the case, model function is affected, and high error rates and low precision levels are generated, as is significant statistical data dispersion. Thus, to predict performance and implement prevention programs for workers, data should be grouped from culturally, politically, socially, and economically similar regions.

Conclusions

A novel approach, the Physical Surface Tension-Neural Net (PST-NN), was proposed in this study to classify psychosocial risk levels among Colombian state-school teachers. Psychosocial, physiological, and musculoskeletal input variables were used to train and test the PST-NN, as a function of four risk level groups (low, medium, high, and very high risk).

The proposed method obtained better classification results than models such as Decision Tree, Naïve Bayes, Artificial Neural Networks, Support Vector Machines, Hill-Climbing-Support Vector Machines, k-Nearest Neighbor-Support Vector Machine, Robust Linear Regression, and Logistic Regression. Indeed, the PST-NN had an average accuracy value of 97.31%, including all the risk level groups and all the subjects in the database.

The results obtained in the prediction of the model demonstrated that the proposed PST-NN approach is applicable for the identification of the psychosocial risk level among Colombian state-school teachers, with high levels of accuracy, and it may contribute as a tool in the generation of guidelines in public health plans, defined by the government.

Supplemental Information

Supplemental Information 1 Consent form: original version.

Click here for additional data file.

Supplemental Information 2 Code

Click here for additional data file.

Supplemental Information 3 Dataset: Psychosocial Risk Colombian Teachers-School

All evaluations of psychosocial patients which had and, didn’t have, any psychosocial. These evaluation patients were used for the dataset modelling and statistical analysis.

Click here for additional data file.

Additional Information and Declarations

Competing Interests

Author Contributions

Ethics

Data Availability

The authors declare there are no competing interests.

Rodolfo Mosquera Navarro conceived and designed the experiments, performed the experiments, analyzed the data, performed the computation work, prepared figures and/or tables, and approved the final draft.

Omar Danilo Castrillón, Liliana Parra Osorio, Tiago Oliveira, Paulo Novais and José Fernando Valencia analyzed the data, authored or reviewed drafts of the paper, and approved the final draft.

The following information was supplied relating to ethical approvals (i.e., approving body and any reference numbers):

The dataset is a self-administered survey by labor psychologist and it was approved by university ethics committee public health at Universidad Nacional de Colombia, sede Manizales (Acta 01, SFIA-0038-17, legal document Mz. ACIOL-009-17, January 18, 2017).

The following information was supplied regarding data availability:

The raw measurements are available in the Supplementary Files.

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
