# Peer review of "Improving classification based on physical surface tension-neural net for the prediction of psychosocial-risk level in public school teachers"

_PeerJ Computer Science, doi:10.7717/peerj-cs.511_

## Round 0.1 · original submission · Major Revisions

Please pay careful attention to the reviewers' comments, which I summarize here.

Both reviewers note that there are many errors in English grammar and the writing should be edited for greater clarity. Please also note that if a section is broken into subsections there should be at least two subsections. These errors make interfere with reading the paper.

Reviewer 1 pointed to the need for more careful scholarship and for citations to accompany some of the statements that were made. Reviewer 1 also suggested comparing artificial neural networks with classical regression, and provided a useful reference.

Please clarify whether the collection and use of data was overseen by an Institutional Review board for approving human subjects research.

Please also follow closely the technical suggestions made by the reviewers. Explain why several different splits were made between training, validation, and test samples. Please also provide a ROC curve.

Reviewer 1 ·

Basic reporting

the introduction is excessively long, scatter, and irregular. The authors should avoid additional descriptions and studies.

Experimental design

The paper does not mention review of this research protocol by an ethics review board before the research was conducted. Obtaining consent is not the same as having an ethics review board vet the study before it is done.

Validity of the findings

no idea.

Additional comments

In this article, the authors have used "the physical surface tension-neural net", which is a nice methodological choice. As well as this is an interesting topic that present novel data. However, I can mark the present work as a major revision as there are a lot of scattering and irregularity during the work. Therefore, the authors are requested to answer and deal with the comments one-by-one and very closely.

Abstract:
1. I think that the abstract is very long.
2. I think that in the abstract authors should avoid using acronyms.
3. the conclusion of the abstract is vague and unrelated and it should be rewritten.

Introduction:
1. the introduction is excessively long, scatter, and irregular. The authors should avoid additional descriptions and studies.
2. It should be noted that from time to time there are some English grammatical errors in the whole manuscript. The manuscript needs to be revised by a native English editor. Of course, although the text benefits from suitable and sophisticated information presented in the sentences, some sentences have not been written correctly, which make them somehow understandable for possible readers and researchers.
3. In introduction the authors made some statements without the use of appropriate references. For example, " Psychosocial risks are stress factors that can alter and unbalance the resources and abilities of the person to manage and respond to the flow of the activity derived from work, negatively affecting physical and psychological health". " Liquid surface tension is defined as the amount of energy necessary to increase surface area of a liquid per unit of area. Surface tension (a manifestation of liquid intermolecular forces) is the force that tangentially acts, per unit of longitude, on the border of a free surface of a liquid in equilibrium, and which tends to contract said surface.", ….
4.You have too many abbreviations in the study; doesn't this make the article difficult to understand by readers and researchers?
5. It's better for authors to express the importance of artificial neural networks compared to classical regression methods:
- Farhadian M, Aliabadi M, Darvishi E. Empirical estimation of the grades of hearing impairment among industrial workers based on new artificial neural networks and classical regression methods. Indian J Occup Environ Med. 2015 May-Aug; 19(2): 84–89. doi: 10.4103/0019-5278.165337.

Material and Methods
1. The method part is not satisfactory. Subsections must be modified as below:
- Description of the samples (teachers)
- Description of model variables.
- Database and Data Pre-processing
- Description of Model construction process:
¬ - Step 1: …
- Step 2 …
- Step 3 …
- Step 4
- Comparison techniques

2. The variables considered has been collected based on what standard or reference? For example, the musculoskeletal symptoms have been collected using what tools? Moreover, there is no reference to the tools and methods of measuring physiological parameters.

3. The authors should add references that support the defined methodology.
4. The paper does not mention review of this research protocol by an ethics review board before the research was conducted. Obtaining consent is not the same as having an ethics review board vet the study before it is done.
5. There could be a more organized flow throughout the manuscript and greater organization of the methods.

Results:
1. The number of tables is very large. Authors can merge a number of tables. It will be very boring for the reader.
2. 2. The authors can add one table and submit the predicted risk level by the PST-NN model in the 1-3 cases of new samples.

Discussion:
1. Overall, discussion should be improved. The authors should avoid to describe the results again. They should only explain, with appropriate support, why the results were obtained. Additional explanations are missing.
2. What do the authors think with ANN and other models? I invite the authors to read, among others, the following article:
Aliabadi M, Farhadian M, Darvishi E. Prediction of hearing loss among the noise-exposed workers in a steel factory using artificial intelligence approach. International Archives of Occupational and Environmental Health volume 88, pages779–787(2015).

Conclusion:
I advise that the conclusion was rewritten based on followed:
“The proposed hybrid method in this investigation, PST-NN, obtained better results than other models for the prediction of the psychosocial risk level in Colombian public-school teachers. The percentage of classification was 97.37% for the PST-NN approach. Generally, the input parameters were selected in accordance with researchers' experience, but in this model, the Physical Surface Tension method performed better in the resolution of parameter optimization, which transformed and reduced from 20 to 2 physical variables, and reduced ANN deficiencies. The results obtained in the prediction demonstrate that the proposed prediction model is applicable for the identification of the psychosocial risk level in Colombian public-school teachers with high levels of accuracy and contribute to the guidelines with the public health plans defined by the government.”

Reviewer 2 ·

Basic reporting

The use of English is unclear and ambiguous. There are some grammatical errors and typos such as:
- ... is to develop a model that support machine learning model to ...
- (Collie, Shapka & Perry, 2012) in a previous study shown how teachers’ perception of their work environment influenced in the levels of: ...
- Prediction of psychosocial risk levels in public school teachers have to fundamental importance, owing to its impact on teacher health.
- According this, the goal were develop a model be able to mathematically relate the psychosocial factors ...
- The review of literature show that models with different applications have been proposed for classification purposes, however, don´t anything ...
- Therefore, it can be argued that such a development is a novel approach that adds to the scientific of literature in this field.
- ...
The authors should re-check and revise carefully. It is necessary to be proofread by a native speaker or editing service.

Literature review is long and some of them are not related to the study. The authors should optimize and re-organize the literature review.

Experimental design

What is "sensibility level"?

The authors should re-organize the methodology section, now it is not easy to follow. Also, there are some missing sections, i.e., there is a section "2.2.1" but not "2.2.2"? Now it is difficult to replicate the methods with this presentation.

Measurement metrics (such as sensitivity, specificity, accuracy) have been used in previously published works such as PMID: 28643394, PMID: 32942564, and PMID: 32613242. Therefore, the authors should refer more works in this description to attract broader readership.

"Classification error" is an useless metric, since we can easily get it by (100 - accuracy).

Why did the authors run the tests 6 times with different ratios of train/val/test?

It is important to have ROC curve and AUC analysis

Validity of the findings

The authors proposed a lot of things in methodology section, but very few information in results section. It did not show clearly the findings and impacts of the results.

Also, the authors should add more discussions.

In Table 6, why did the authors only report the training performance? Where is validation and testing performance?

According to Fig. 6, the model achieved the best performance at about 100 iterations, why did the authors not stop their training process at this cut-off level? It the authors continue to train, the model will lie to overfitting problem.

How did the authors deal with overfitting problem, since the performance between training and validation/testing are different?

Additional comments

No comment

---

## Round 0.2 · Major Revisions

Please see comments from reviewers, especially Reviewer 2, who has serious concerns that were not addressed in the previous revision.

In your revision, please pay special attention to the issue of overfitting. Figure 6 suggests there was considerable overfitting (the mean squared error on the training sample is much less than on training and validation). This should be acknowledged.

Tables 6 and 7 seem to lump training, validation, and test samples all together. This is wrong, especially when there is overfitting. Results on the test set should be reported separately in these tables so the reader can understand how it performs on data the algorithm has not previously seen.

I find the remarks on cross-validation very confusing. You say 5-fold cross-validation was used with three groups used for training and one as the test. This is an incorrect use of the term 5-fold cross-validation. In 5-fold cross-validation, the training and validation data is split into 5 subsets. The model is fit 5 times. Each time it is fit on 4 subsets and evaluated on one holdout subset. A good explanation can be found at this link: https://machinelearningmastery.com/k-fold-cross-validation/

As Reviewer 2 states, cross-validation is not a way to prevent overfitting of model parameters. The purpose of cross-validation is to give an unbiased estimate of the true prediction error. It allows you to evaluate the degree of overfitting, but it does not prevent overfitting. On the other hand, cross-validation can be used to prevent overfitting of algorithm hyperparameters, as described here: https://elitedatascience.com/overfitting-in-machine-learning

The paper states: "the data set was divided into training, validation and test groups with different data division percentages and 5-fold cross-validation, as shown in Table 4." Table 4 says nothing about cross-validation. If you really used 5-fold cross-validation, then the validation set should be 1/4 the size of the training set, which is not the case in Table 4. Also, as noted by Reviewer 2, you have not explained why you repeated the process with different sizes of training, validation, and test samples.

Please present a detailed explanation of the procedure used for fitting. Please give enough detail that the reader can verify that cross-validation was correctly applied. Please also make sure your tables do not lump results on training, validation, and test samples all together. Please also address the other concerns of Reviewer 2.

Reviewer 1 ·

Basic reporting

nothing to add

Experimental design

nothing to add

Validity of the findings

nothing to add

Additional comments

In my opinion, many of the comments have remained unchanged.
particularly in method and material, the authors should be modified the subsections the order as below:
- Description of the samples (teachers)
- Description of model variables.
- Database and Data Pre-processing
- Description of Model construction process:
¬ - Step 1: …
- Step 2 …
- Step 3 …
- Step 4
- Comparison techniques

Reviewer 2 ·

Basic reporting

No comment

Experimental design

No comment

Validity of the findings

No comment

Additional comments

Thanks for addressing my previous comments. Although some efforts have been done, the authors have not addressed well in most of my comments. Therefore, I do not suggest to accept this manuscript for publication. Some comments are still not answered well as follows:

1. Literature review is long and some of them are not related to the study. The authors should optimize and re-organize the literature review.

2. Why did the authors not have section 2, but they have sub-sections 2.1, 2.2., ...?

3. Measurement metrics (such as sensitivity, specificity, accuracy) have been used in previously published works such as PMID: 28643394, PMID: 32942564, and PMID: 32613242. Therefore, the authors should refer more works in this description to attract broader readership.

4. Why did the authors run the tests 6 times with different ratios of train/val/test? ==> Table 7 did not show any reason that the authors used different ratios of different test times.

5. The authors proposed a lot of things in methodology section, but very few information in results section. It did not show clearly the findings and impacts of the results. ==> I even don't understand the answer from the authors in this question. (We believe that the evaluator confuses the methods of previous studies with this one and the results.
The results of the proposed method shown are those sufficient and necessary to evauate a new classification method, regarding both stability and overfitting. However, as the evaluator suggested, the ROC metric was added to guarantee the adequate per-formance of the algorithm.
)

6. In Fig. 6, the best performance might come at epoch of 100, why did the authors need to run to 1000 epochs?

7. The authors mentioned "To prevent overfitting, the preprocessed dataset was split in 5-fold cross validation", but I think it is not a solution to prevent overfitting. This is a normal evaluation method for all machine learning problems. I'd like to ask whether the authors applied some other techniques to avoid overfitting in their models. According to the results, I think the models contained a lot of overfitting.

---

## Round 0.3 · accepted · Accept

The reviewers agree that your revisions have met their concerns.

Reviewer 1 ·

Basic reporting

There is nothing special in this area.

Experimental design

There is nothing special in this area.

Validity of the findings

There is nothing special in this area.

Additional comments

There is nothing special in this area.

Reviewer 2 ·

Basic reporting

No comment.

Experimental design

No comment.

Validity of the findings

No comment.

Additional comments

No comment.